# Using Proximal Hyperspectral Sensing to Predict Herbage Nutritive Value for Dairy Farming

**Federico N. Duranovich** [1,*]**, Ian J. Yule** [2]**, Nicolas Lopez-Villalobos** [1]**, Nicola M. Shadbolt** [1]**, Ina Draganova** [1] **and Stephen T. Morris** [1]

[1] School of Agriculture and Environment, College of Sciences, Massey University, Private Bag 11-222, Palmerston North 4442, New Zealand; N.Lopez-Villalobos@massey.ac.nz (N.L.-V.); N.M.Shadbolt@massey.ac.nz (N.M.S.); I.Draganova@massey.ac.nz (I.D.); S.T.Morris@massey.ac.nz (S.T.M.)

[2] Massey AgriFood Digital Lab, School of Food and Advanced Technology, College of Sciences, Massey University, Private Bag 11-222, Palmerston North 4442, New Zealand; I.J.Yule@massey.ac.nz

\* Correspondence: F.N.Duranovich@massey.ac.nz

**Abstract:** This study focuses on calibrating and validating models for hyperspectral canopy reflectance data that are useful to predict the nutritive value of ryegrass-white clover mixed herbage available to the grazing cow. Hyperspectral measurements and herbage cuts were collected from 286 sampling plots from a dairy farm from July 2017 to May 2018. Hyperspectral data were pre-treated by applying a Savitzky-Golay filter followed by a Gap-segment derivative algorithm. Herbage samples were analyzed for determination of herbage nutritive value traits, digestible organic matter in dry matter (DOMD), metabolizable energy (ME), crude protein (CP), neutral detergent fiber (NDF) and acid detergent fiber (ADF). Partial least squares regression was performed to calibrate the spectra against the five nutritive value traits. Results indicate that accuracy was moderately high for the CP model ($R^2 = 0.78$) and moderate for the DOMD, ME, NDF and ADF models ($0.54 < R^2 < 0.67$). The possibility of being able to use proximal sensing for the estimation of herbage nutritive value in the field could potentially contribute to more efficient grazing management with potential economic benefits for the farm business.

**Keywords:** proximal hyperspectral sensing; herbage nutritive value measurement; grazing management; partial-least squares regression

---

## 1. Introduction

Research suggests that improved knowledge of herbage mass availability of the farm can increase dairy farm profits by increasing the precision of daily feed allocation [1]. However, allocating feed with a sole focus on herbage mass measurement may not result in the most optimal allocation decisions. Feed allocation decisions aim at achieving cow feed intakes that would deliver desirable farm performance targets. However, in addition to mass, feed intake depends on the energy and nutrient content available in herbage (i.e., its nutritive value, NV) [2,3]. The decision on the amount of herbage to allocate to cows may therefore differ if herbage resources vary in their NV [3]. If differences in herbage NV are not considered in the feed allocation process, then differences between actual and expected herbage intakes should be expected. At any grazing event, an actual dry matter intake that differs from expected can result in actual animal performance, animal response to supplements and post-grazing residual being different from expected targets [4,5]. In the long term, consistently defoliating herbages at suboptimal post-grazing residual heights will result in increased feed wastage and reduced herbage production, herbage persistence, herbage utilization, NV and milk production at the farm level, ultimately reducing the potential profitability of the farm system [1,6,7].

Rapid, objective herbage NV measurement has been identified as an important opportunity to improve grazing management on pasture-based dairy farm systems [8]. Nevertheless, the lack of measurement tools available for their use on farms has limited the possibility of taking advantage of such opportunity. Objective measurement of herbage NV has traditionally involved grab sampling representative samples of herbage at the height grazed by the cows and sending these to a laboratory for analysis [9]. Analysis of samples usually involves determination of digestible organic matter in dry matter (DOMD), metabolizable energy (ME), crude protein (CP), neutral detergent fiber (NDF) and acid detergent fiber (ADF) [2]. Wet chemistry or near-infrared spectroscopy (NIRS) are the most common laboratory techniques used for the determination of herbage NV [10,11]. The whole process of collecting, preparing and analyzing samples is expensive and time-consuming, making it impractical for their use in rapid decision-making. Field assessment of herbage quality indicators such as leaf stage of grass species [12], leaf to stem ratio, stage of growth, species composition and proportion of legume in the sward, can be used as surrogate measures of herbage NV [13]. Although useful, these indicators fail to provide the actual objective measures required for precise diet formulations, because other factors such as soil moisture and fertility also affect herbage NV [2]. Alternatively, mechanistic models capturing the causal relationships between herbage growth and structural processes and herbage NV can offer a more functional approach that can be used for predicting herbage NV [14]. Despite being a powerful research tool, mechanistic models might not be accurate to determine herbage NV under the varying conditions of commercial farms, for which a dedicated tool for measuring herbage NV in the field would be required.

Increasing attention is being placed on sensing technology as a suitable tool for delivering rapid and objective measures of herbage NV [8]. Spectral signatures of herbage canopies have unique features that are useful to characterize biochemical properties associated with their NV [15,16]. For instance, it is well established that canopy reflectance in the visible region of the electromagnetic spectrum is strongly determined by chlorophyll pigments [17,18]. Since most nitrogen in plant tissue is contained in chlorophyll–protein complexes, strong relationships between canopy reflectance in the visible wavelengths and CP content have been established [19,20].

Progress in sensor development has resulted in increased spectral and spatial resolutions and the possibility of studying herbage NV with greater accuracy and at different spatial and temporal scales [16,21]. Sensors mounted on satellites or unmanned aerial vehicles have been valuable for studying the spatial variation of NV of sown pastures and natural grasslands with detail and at large scales [15,22–24]. Although useful for studying the spatial variation of vegetation, the challenges imposed by the weather [25], cloud distortive effects [23] and the unavailability of satellite images on a regular basis [23] pose a limit to the temporal scale at which remote tools can be used. Alternatively, proximal sensors offer a flexible alternative to the study of phenomena requiring regular and frequent spectral measurements.

Proximal sensors can be carried by hand or mounted on vehicles for speed capability [26] and used in conjunction with active lighting systems to allow independence of ambient light [27,28]. A number of researchers have studied the relationships between canopy spectral features and herbage NV using proximal optical [29], multispectral [30,31] and hyperspectral [20,32–34] sensors with varying success. Overall, empirical models based on high-resolution hyperspectral sensor data are more accurate than those using multispectral sensing, and accuracies improve if an active lighting system is used. Although much of the research on the topic has focused on establishing empirical relationships in relatively well-controlled conditions, recent research [20,34] demonstrated that proximal hyperspectral sensing is also suitable for measuring herbage NV in the field. These advances show the potential of sensing technology for their use in commercial farm management, but despite success, the work completed so far has not been developed into a commercial solution.

To date, all of the research on the topic has focused on finding relationships between the spectra of canopies and the NV of tissue in the whole herbage profile [20,32,34], without considering that NV decreases with canopy depth [9,35,36] and that only a limited portion of the vertical profile of

herbage is made available to the grazing cow [7]. Consequently, the traditional sampling strategy used to characterize herbage NV from canopy spectra is not representative of good grazing management practice. On the other hand, although proximal hyperspectral sensors measure the first surface they sense [27], there is evidence that, even at full closure, canopy spectral signatures of grass herbage can also be influenced by the lower strata not grazed by cows [37]. This could potentially affect the use of proximal hyperspectral sensing for predicting of herbage NV for grazing management, as sensed but ungrazeable material at the bottom of the canopy might affect the calibration of the instrument. Research is required to determine the capability of proximal hyperspectral sensing to measure herbage NV of the portion of herbage that is made available to the grazing cow. If found useful, proximal hyperspectral sensors can provide a rapid NV measurement tool that could be useful to allocate herbage and supplements to cows with greater precision, with positive consequences to the overall efficiency of the farm system.

This study aims to calibrate and validate models for hyperspectral canopy reflectance data that are useful to determine herbage NV traits DOMD, ME, CP, NDF and ADF from the vertical portion of the sward that should be made available to lactating dairy cows in accordance to a sampling strategy that reflects good grazing management practice.

## 2. Materials and Methods

### 2.1. Study Site

This study was conducted at Dairy 1, a dairy farm owned by Massey University and located in Palmerston North, New Zealand (latitude = −40°22′35.1′′, longitude = 175°36′51.1′′). The mean annual rainfall from 2002 to 2018 at the location was 968 mm, mean annual temperature is 13.3 °C and mean low and high temperatures are 8.9 and 17.7 °C, respectively [38]. Herbage resources on the farm are mostly composed of perennial ryegrass (*Lolium perenne* L.) and white clover (*Trifolium repens* L.) mix, with some herbages also including red clover (*Trifolium pratense*) as part of the mix. Weeds such as buttercup (*Ranunculus* spp.) and annual poa (*Poa annua*) and herbs such as chicory (*Cichorium intybus* L.) and plantain (*Plantago lanceolata*) are also likely to be found but as a small proportion of the sward. Farm soils comprise a complex assemblage of free-draining alluvial soils including Rangitikei Loamy Sand, Manawatu Fine Sandy Loam, Manawatu Sand Loam/Gravelly phase, Manawatu Mottled Silt Loam and Karapoti Brown Sandy Loam, with these soils being well drained and naturally fertile.

### 2.2. Canopy Spectral Measurements

Canopy spectral measurements were collected every two to three weeks from 31 July 2017 to 10 May 2018 from paddocks at pre-grazing stage (herbage dry matter (DM) mass ≥ 2600 kg DM/ha) using an ASD FieldSpec 4 High-Resolution spectroradiometer (Analytical Spectral Devices Inc., Boulder, CO, USA). The spectroradiometer acquires spectra in a wavelength range from 350 to 2500 nm and has a spectral resolution of 3 nm in the visible near-infrared (VisNIR, 350–1000 nm) and of 8 nm in the near-infrared and shortwave infrared (NIR-SWIR, 1001–2500 nm) region of the spectrum. The spectral sampling interval of the instrument is factory set at 1.4 and 1.1 nm for the VisNIR and NIR-SWIR wavelengths, respectively. To simplify data analysis, wavelength units along the spectrum were standardized to 1 nm with the user interface software ASD RS$^3$ (Analytical Spectral Devices Inc., Boulder, CO, USA).

Noise caused by changes in daylight and wind conditions was minimized by using the instrument in combination with a Canopy Pasture Probe (CAPP) system [27]. The CAPP consists of an inverted handled black plastic bin with a 50 watt tungsten-quartz-halogen lamp (ASD Inc., Boulder, CO, USA) attached on top. The CAPP lamp was powered by a 12 v, 8000 mA lithium polymer battery, which ensured a stable source of light at maximum intensity for a day of fieldwork. Factory-calibrated radiance units were converted to reflectance units by calibrating the instrument against a clean

ceramic white tile that was used as a reflectance standard of 100% light reflectance (i.e., R = 1) [27]. The field-of-view of the spectroradiometer is 25°, resulting in a 316 cm$^2$ measured circular surface area.

Hyperspectral canopy reflectance measurements were collected from 286 plots that were situated in the field so as the maximum range of herbage quality conditions was covered. A sampling plot consisted of the area delimited by a 50 × 50 cm wooden quadrat within which canopy reflectance measurements were acquired. To maximize spectral characterization of the herbage canopy within the area delimited in each sampling plot, two measurements were acquired from three adjacent points each (i.e., measurement points), with six spectral measurements per sampling plot in total. Spectral measurements acquired from each sampling plot were averaged to obtain a single canopy signature per sample.

### 2.3. Herbage Cuts

After canopy spectral measurements in each sampling plot were acquired, herbage was cut to 4 cm in height using hand electric grass clippers and with the aid of a sward stick for height determination. The decision on the cutting height was made upon agreed principles of efficient, profitable grazing management for pasture-based dairy systems [7]. After cutting, herbage samples were stored in labelled clean plastic bags in a polystyrene box with freeze pads in order to avoid heat deterioration of samples. After fieldwork, samples were weighed and oven dried at 70 °C for 48 h. Dried herbage samples were ground to pass a 1 mm sieve and stored in individual sealed plastic bags in a dark dry place for further determination of NV.

### 2.4. Determination of the Nutritive Value of Herbage Samples

Dried and ground herbage samples were analyzed for ME (MJ/kg DM) and percentages of DOMD, CP, NDF and ADF in DM using a benchtop NIRS [10]. The accuracy of the NIRS technique for determining the NV of dried and ground herbage samples is provided in Table 1.

**Table 1.** Accuracy of bench-top near-infrared spectroscopy (NIRS) analysis for nutritive value (NV) of dried and ground herbage samples.

| NV Trait | R$^2$ | RMSE | RPE | Bias | RPD |
|:---:|:---:|:---:|:---:|:---:|:---:|
| ME | 0.92 | 0.43 | 4.04 | 0.02 | 3.46 |
| CP | 0.94 | 1.13 | 6.34 | 0.08 | 4.27 |
| NDF | 0.87 | 2.95 | 5.76 | 0.70 | 2.86 |
| ADF | 0.76 | 2.27 | 8.24 | −0.03 | 1.84 |
| DOMD | 0.95 | 1.68 | 2.93 | −0.25 | 4.55 |

ME = metabolizable energy, CP = crude protein, NDF = neutral detergent fiber, ADF = acid detergent fiber, DOMD = digestible organic matter in dry matter, RMSE = root mean square error, RPE= root mean square error, RPD = ratio of prediction to deviation.

### 2.5. Outlier Detection

A principal component analysis (PCA) was performed on canopy reflectance data using the 'prcomp' function implemented in RStudio software (version 1.2.5019, RStudio Team, Boston, MA, USA). Principal component scores resulting from the PCA were used to calculate the Mahalanobis distance for each sample with the function 'Moutlier' available in the package 'chemometrics' for RStudio (chemometrics Version 1.4.2, R Package). A sample was assumed an outlier if the Mahalanobis distance value of the sample exceeded the 99.7% percentiles. Using this threshold, 14 of the 286 samples were considered outliers and excluded from further analyses.

### 2.6. Spectral Data Pre-Treatment

Processes of transformation and signal processing were used to reduce abnormalities in the spectral measurements. Abnormalities across spectra might occur at random or systemically due to instrument

internal factors such as differences in calibration among detectors (Analytical Spectral Devices Inc., Boulder, CO, USA) or external factors such as light leakage, background noise or excess of humidity during data collection in the case of field sampling [27]. Pre-treatment enhances absorbance features of spectra by reducing the incidence of abnormalities and improve repeatability of the modelling method, model robustness and accuracy [39].

The first step of pre-treatment was to reduce the signal gaps between the domains of the detector arrays by applying a splice correction gap of five using VisualSpecPro software (VisualSpecPro Version 6.2, Analytical Spectral Devices Inc., Boulder, CO, USA). Data at both ends of the spectrum (<500 nm and >2400 nm) was excluded due to increased noise most likely caused by ambient light leaking into the CAPP [20], low signal-to-noise ratio or step problems associated with movement of the fiber optic bundle of the instrument [27]. Further transformation involved converting spectra from reflectance (R) units to absorbance log (1/R) units to reduce non-linearity [10].

Residual noise in the converted spectra caused by additive and multiplicative scattering effects unrelated to the chemical nature of samples [40] was treated by applying the 'gapDer' function available in the package 'prospectr' for RStudio software (prospectr Version 0.13, R Package). This function applies the Savitzky-Golay filtering followed by a Gap-segment derivative algorithm to the data. The window size of the smoothing filter was set at 45 and the gap-derivative function calculated for a first-order derivative of segment size 20. The effects of variation in baseline shift and curvilinearity were corrected by de-trending the spectrum by fitting a second-order polynomial to the signal and then subtracting it [41]. The final transformation step was to standardize the spectra by centering each wavelength to a zero mean and scaling it to a variance of one. From now on, the spectral data that resulted from the spectra pre-treatment described above will be referred to as the first derivative of absorbance (FDA).

## 2.7. Calibration Model Development

Hyperspectral calibration models were developed for predicting the selected NV traits from canopy spectral data using the partial least-squares (PLS) regression modelling technique implemented with the 'PLS' package for RStudio software (PLS Version 2.7.2, R Package). Calibration models can be conceptualized using the following general regression equation:

$$y = x_1 + b_2 x_2 + \ldots + b_n x_n + e \tag{1}$$

where: y is the NV trait response variable of interest, $x_1$ to $x_n$ are independent latent variables obtained from reduced wavelengths, $b_1$ to $b_n$ are the partial regression coefficients and e is the residual error that is not explained by the model.

The PLS technique reduces the high number of wavelengths in hyperspectral data to a few uncorrelated latent variables (LVs) and then regresses the scores of the LVs that account for the most variance to the response variable of interest [42]. The optimal number of LVs to retain in the models was selected as that yielding the minimum root mean square error of leave-one-out cross-validation.

The contribution of each wavelength to the predictive capability of the models was interpreted by calculating the variable importance in projection (VIP) [43] and PLS regression coefficients (RC) [44] using the VIP and RC functions available in the package 'plsVarSel' for RStudio software (plsVarSel, Version 0.9.6, R Package). The VIP measures the importance of each predictor variable (i.e., wavelength) based on a model with a defined number of factors (LVs) [43], whilst the PLS regression coefficients are a single measure of association between each wavelength and the response variable [44].

## 2.8. Model Accuracy Assessment

Model accuracy was defined as the overall distance between predicted and observed values and was assessed by randomly splitting the spectral dataset into two independent sets: training and validation sets. The training set comprised 80% of all data and was used to calibrate the models.

The validation set comprised the remaining 20% of all data and was used to test the capability of calibrated models of predicting a set of unknown samples. Metrics used for assessing goodness of fit measures of calibrations models are presented in Table 2.

**Table 2.** Metrics of goodness of fit measures of the calibration models.

| Metric | Equation | |
| --- | --- | --- |
| Coefficient of determination | $RMSE = \dfrac{\sum_{i=1}^{n} (\hat{y} - \bar{y})^2}{\sum_{i=1}^{n} (y - \bar{y})^2}$ | (2) |
| Root mean square error | $RMSE = \sqrt{\dfrac{\sum_{i=1}^{n} (\hat{y} - y)^2}{n}}$ | (3) |
| Relative prediction error | $RPE = \dfrac{RMSE}{\bar{y}} \times 100$ | (4) |
| Bias | $Bias = \dfrac{1}{n} \sum_{i=1}^{n} (\hat{y} - y)$ | (5) |
| Ratio of prediction to deviation | $RPD = \dfrac{SD\ (y)}{RMSE}$ | (6) |

$\hat{y}$ = predicted value, y = measured value, $\bar{y}$ = mean of measured values, n = number of observations, SD = standard deviation of measured values.

The coefficient of determination and the root mean square error (RMSE) are the most common metrics used for model fit assessment. The $R^2$ (Equation (2)) indicates the proportion of the variance in the reference data that is accounted for by the regression model. RMSE (Equation (3)) is the standard deviation of the residuals (prediction errors) and represents an absolute measure of model accuracy. The relative prediction error (RPE) is the RMSE expressed as a percentage of the mean of the measured values (Equation (4)), resulting in a standardized measure that is useful to compare models predicting responses of different magnitudes and/or units. Bias (Equation (5)), is the mean difference between the predicted and measured values and is useful to identify systemic errors in the models. Negative bias values indicate a generalized sub-estimation by the models, while positive values indicate over-estimation. Finally, ratio of prediction to deviation (RPD) (Equation (6)) is the factor by which prediction accuracy increases compared with using the mean of measured values.

## 3. Results

### 3.1. Descriptive Statistics of Reference Nutritive Values and Spectral Data

The NV of herbage samples used for training and validating calibration models exhibited similar range, mean, SD and CV values (Table 3). Crude protein was the most variable NV trait while DOMD was the least variable NV trait.

**Table 3.** Accuracy of benchtop near-infrared spectroscopy (NIRS) analysis for nutritive value (NV) traits of dried and ground herbage samples.

| NV Trait | Dataset | N | Range | Mean | SD | CV |
| --- | --- | --- | --- | --- | --- | --- |
| ME | Training | 220 | 8.27–12.07 | 10.72 | 0.74 | 0.07 |
| | Validation | 52 | 7.66–11.99 | 10.72 | 0.82 | 0.08 |
| CP | Training | 220 | 5.58–25.65 | 17.88 | 3.69 | 0.21 |
| | Validation | 52 | 7.04–23.88 | 17.52 | 4.50 | 0.26 |
| NDF | Training | 220 | 30.28–55.52 | 40.42 | 4.77 | 0.12 |
| | Validation | 52 | 31.14–57.93 | 40.39 | 5.38 | 0.13 |
| ADF | Training | 220 | 14.81–28.66 | 21.32 | 2.69 | 0.12 |
| | Validation | 52 | 15.57–31.51 | 21.17 | 3.03 | 0.14 |
| DOMD | Training | 220 | 55.21–69.86 | 64.80 | 2.54 | 0.04 |
| | Validation | 52 | 53.37–69.21 | 64.78 | 3.01 | 0.05 |

ME = metabolizable energy (MJ/kg DM), CP = crude protein (%DM), NDF = neutral detergent fiber (%DM), ADF = acid detergent fiber (%DM), DOMD = digestible organic matter in dry matter (%DM), DM= dry matter, CV = coefficient of variation.

Dispersion of NV was sufficient to establish significant interrelationships among traits (Table 4).

**Table 4.** Correlation matrix of herbage nutritive value (NV) traits of training and validation datasets.

| NV Trait | Training | | | | | Validation | | | | |
|---|---|---|---|---|---|---|---|---|---|---|
| | ME | CP | NDF | ADF | DOMD | ME | CP | NDF | ADF | DOMD |
| ME | 1.00 | | | | | 1.00 | | | | |
| CP | 0.31 *** | 1.00 | | | | 0.33 * | 1.00 | | | |
| NDF | −0.76 *** | −0.28 *** | 1.00 | | | −0.86 *** | −0.43 ** | 1.00 | | |
| ADF | −0.75 *** | −0.21 *** | 0.90 *** | 1.00 | | −0.83 *** | −0.35 * | 0.88 *** | 1.00 | |
| DOMD | 0.75 *** | 0.30 *** | −0.87 *** | −0.77 *** | 1.00 | 0.87 *** | −0.32 * | −0.93 *** | −0.83 *** | 1.00 |

\* Significant at $p < 0.05$, \*\* Significant at $p < 0.01$, \*\*\* Significant at $p < 0.001$, ME = metabolizable energy, CP = crude protein, NDF = neutral detergent fiber, ADF = acid detergent fiber, DOMD = digestible organic matter in dry matter.

Significant and positive relationships ($p < 0.05$) were observed between fiber traits and among DOMD, CP and ME, and with traits in these two groups being negatively associated to each other (Table 4). The strength of the relationships varied among traits, with stronger correlations being observed among DOMD, ME and NDF, and weaker correlations between CP and NDF, ADF, ME and DOMD.

Canopy reflectance mean values (Figure 1a) exhibited two major peaks in NIR wavelengths ranging from 750 to 1350 nm and SWIR wavelengths from 1500 to 1850 nm, and two relatively smaller peaks in wavelengths in the visible (520–570 nm) and far-SWIR (2150–2250 nm) regions of the spectrum. Variation of canopy reflectance was higher in wavelengths ranging from 630 to 690 nm, from 1450 to 1550 nm and from 1900 to 2400 nm (Figure 1a). Spectra variation after pre-treatment (Figure 1b) was higher in the waveband centered at 1050 nm. High variation was also observed in the wavebands centered at 700, 1400 and 1800 nm.

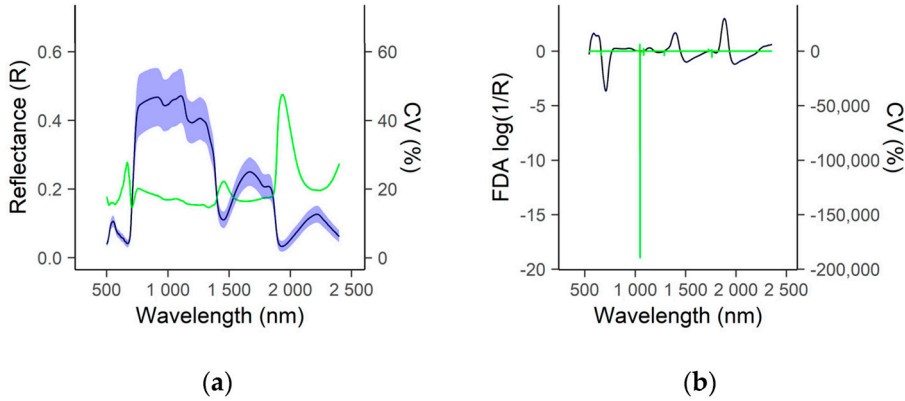

(**a**)　　　　　　　　　　　　　(**b**)

**Figure 1.** Wavelength mean values and coefficients of variation (CV) of (**a**) reflectance and (**b**) first derivative of absorbance (FDA) of herbage canopies. The black solid line is the wavelength mean value and the blue shaded area represents the data within one standard deviation above and below the mean. The green solid line is the wavelength CV (n = 272).

*3.2. Prediction of Herbage Nutritive Value Using Proximal Canopy Spectra*

3.2.1. Model Accuracy

The accuracy of PLS regression models was satisfactory (Table 5). Accuracy values for the trained and the validation data were consistent. Low RMSE, RPE and bias values and high $R^2$ and RPD values indicated that the spectra of canopies were useful to predict the NV of the portion of the herbage that should be made available to the grazing cow. However, model accuracy varied depending on the NV trait modelled.

**Table 5.** Accuracy of partial least-squares regression calibration models built for determining the nutritive value (NV) traits of herbage available for grazing from canopy spectral measurements using the training and validation datasets.

| Dataset | NV Trait | $R^2$ | RMSE | RPE | Bias | RPD |
|---------|----------|-------|------|-----|------|-----|
| Training | ME | 0.67 | 0.42 | 3.96 | $6.90 \times 10^{-16}$ | 1.89 |
| | CP | 0.78 | 1.76 | 9.87 | $-5.21 \times 10^{-16}$ | 2.43 |
| | NDF | 0.54 | 3.33 | 8.25 | $-3.44 \times 10^{-15}$ | 1.54 |
| | ADF | 0.55 | 1.80 | 8.46 | $3.15 \times 10^{-16}$ | 1.48 |
| | DOMD | 0.62 | 1.65 | 2.55 | $-4.68 \times 10^{-15}$ | 1.64 |
| Validation | ME | 0.59 | 0.52 | 4.88 | $4.13 \times 10^{-2}$ | 1.46 |
| | CP | 0.77 | 2.05 | 11.73 | $4.41 \times 10^{-1}$ | 1.84 |
| | NDF | 0.55 | 3.23 | 7.96 | $-2.15 \times 10^{-1}$ | 1.50 |
| | ADF | 0.56 | 1.98 | 9.35 | $-1.13 \times 10^{-1}$ | 1.27 |
| | DOMD | 0.58 | 1.60 | 2.47 | $1.72 \times 10^{-1}$ | 1.60 |

ME = metabolizable energy, CP = crude protein, NDF = neutral detergent fiber, ADF = acid detergent fiber, DOMD = digestible organic matter in dry matter.

Results show that CP was predicted with the highest accuracy among all NV traits ($R^2$ = 0.78 and RPD = 2.43), with the accuracies of the remaining models being relatively lower (0.54 < $R^2$ < 0.67 and 1.48 < RPD < 1.89).

### 3.2.2. Wavelength Contribution to the Predictive Capability of the Calibration Models

The contribution of each wavelength to the predictive ability of the calibration models is illustrated by the VIP scores and RC in Figure 2. Shaded areas indicate wavelengths above a VIP threshold of 1 and are of high importance for predicting the response variable [45]. Regression coefficients above or below zero indicate positive or negative relationships between a wavelength and the predicted NV trait of interest, respectively.

A number of wavelengths in the VIS (540–700 nm), NIR (750–800 nm, 900–1000 nm and 1100–1400 nm) and SWIR (1820–1880 nm and 2125–2300 nm) regions of the spectrum were important and common predictors across models.

Greater similarities in the VIP patterns were observed between the fiber calibration models, and between the fiber and DOMD and ME models. Wavelengths in the 2200 to 2240 nm range were particularly important for the calibration of ADF, NDF, DOMD and ME, with the first derivative of absorbance in these wavelengths being positively related with fibers but negatively related with DOMD and ME. Other relevant wavelengths associated with these calibration models were located in the 620 to 700 nm, 910 to 990 nm, 1040 to 1180 nm, 1250 to 1325 nm, 1425 to 1480 nm, 1920 to 1980 nm and 2330 to 2360 nm ranges. The 1820 to 1860 nm range was of particular relevance for ADF and NDF but not for DOMD or ME, while wavelengths in the range from 2020 to 2100 nm were important for predicting ME, DOMD and CP but not fiber.

Wavelengths ranging from 540 to 680 nm and from 1050 to 1380 nm were of high importance in the determination of CP. Likewise, absorbance in SWIR wavelengths ranging in the 2140 to 2300 nm region was also an important determinant of CP.

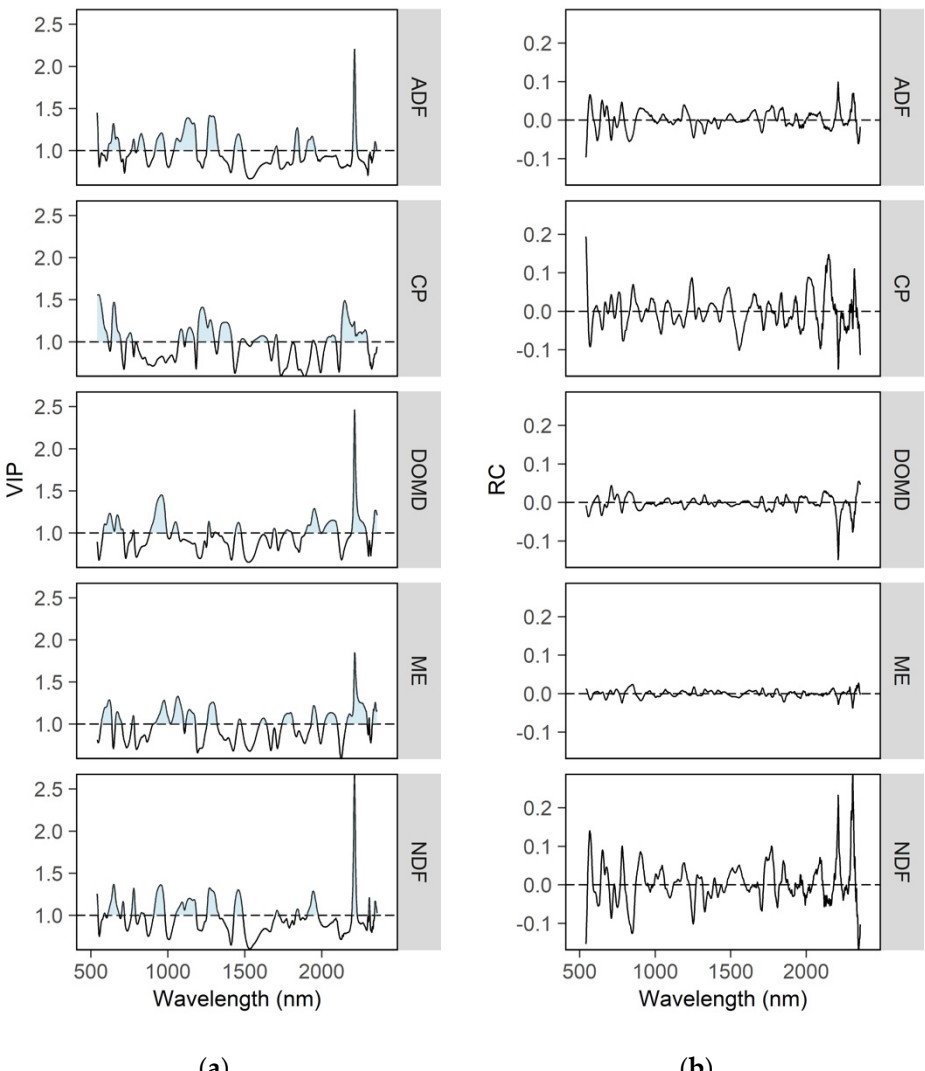

**Figure 2.** Variable of importance in projection (VIP) scores (**a**) and regression coefficients (RC) (**b**) of canopy spectral calibration models developed for the determination of herbage nutritive value traits. ME = metabolizable energy, CP = crude protein, NDF = neutral detergent fiber, ADF = acid detergent fiber, DOMD = digestible organic matter in dry matter.

## 4. Discussion

### 4.1. Representativeness of the Data Used for Building the Calibration Models

Mean herbage nutritive values of the samples used in the development and validation of canopy spectral calibration models were similar to benchmark values commonly used in the dairy industry [46]. DairyNZ [46] reports reference values for ryegrass-based herbages of 10.7 MJ/kg DM for ME, 16.8% for CP and 47.3% for NDF. Similar values for ME, CP, DOMD, NDF and ADF are found in other studies [47,48]. These similarities indicate that the samples collected were an adequate standard with reference to the NV of the herbage available for grazing management, and therefore, suitable for the purpose of this study. Although the correlations between CP and any of the other NV traits measured were not very strong, correlations found here were consistent with previous research [48] and are explained by the relative contribution of the chemical constituents of plant cells present on herbage samples and their relationship with the different measured herbage NV traits. Likewise, canopy reflectance (Figure 1a) exhibited the typical pattern associated with vegetation spectral signatures [16]. High variation of absorbance in the waveband centered at 1050 nm could potentially

be explained by differences of herbage water content associated with samples collected throughout the year [49]. Curran [49] described that vegetation water content can influence spectra in the wavelengths ranging from 800 to 1300 nm.

### 4.2. Predictive Capability of Calibration Models

Consistency of accuracy for calibration and validation datasets across models indicated that the models were robust and therefore useful for predicting new samples [50]. This finding was of relevance to this study since it signified that proximal hyperspectral sensing could potentially be used in an array of field conditions with confidence, with this attribute being a desirable characteristic of herbage measuring tools for farmers [51]. However, as expected, the accuracies of the calibrations developed from field spectral measurements were much lower than the threshold values used in benchtop NIRS spectroscopy of dried, milled and more homogeneous samples [52]. In NIRS-determined laboratory analyses for agricultural products, predictions are deemed 'Inadequate' with $R^2 < 0.70$ and RPD < 1.75. According to these threshold values, only CP was determined with some degree of usefulness. Nevertheless, other authors [53] suggest that field measurements reduce prediction accuracy of NIRS models and so therefore, accuracies lower than laboratory reference standards could also indicate "good results".

There were inconsistencies among the various field studies linking herbage NV and the spectra of herbage canopies [20,32,34]. Our models were able to predict CP, ME, NDF and ADF with lower $R^2$ values than those of Pullanagari et al. [20] ($R^2 = 0.82$, $R^2 = 0.83$, $R^2 = 0.75$, $R^2 = 0.82$ for CP, ME, NDF and ADF, respectively). The $R^2$ value obtained for CP in this research was higher than Adjorlolo et al. [34] ($R^2 = 0.51$) and Kawamura et al. [32] ($R^2 = 0.46$). Results for the fiber models were slightly lower to the models reported by Adjorlolo et al. [34] ($R^2 = 0.60$ for NDF and ADF) but higher than the NDF model by Kawamura et al. [32] ($R^2 = 0.37$) and lower than the ADF model ($R^2 = 0.65$). Despite the difference in the sampling method and differences in the pre-treatment of spectral data, all the models were developed using PLS on the first derivative of canopy absorbance. A major factor influencing model accuracy among studies (and between the models in this study) seemed to be associated with the variability of the reference NV data used in the calibrations, with better performing models being associated with higher coefficients of variation for any of the NV traits considered.

The effect of the variability of the dataset used on model accuracy can be illustrated by the following observation: Pullanagari et al. [20] were able to predict ME with a RMSE of 0.46 MJ/kg DM and a RPD of 2.46 using training data of mixed herbage with a SD of 1.16 MJ/kg DM, while this study reports a similar RMSE (0.42 MJ/kg DM) but a lower RPD (1.89) using a less variable dataset (SD = 0.72 MJ/kg DM). Because RPD is calculated as the ratio between RMSE and SD (Equation (6) in Table 2) and there were similar predictive errors between the models, the higher RPD found by Pullanagari et al. [20] compared to this study is likely to be associated with the higher variability of their dataset rather than in the methods used. Similar observations about the influence of the variability of the dataset on model performance were made by other authors [33,53]. The small range of variability within the data collected was most likely associated with the sampling strategy chosen, and thus the objective of this study. Because modelling of spectra aimed at characterizing herbage NV of herbage to allocate to grazing cows (i.e., herbage at pre-grazing stage), the variability of NV was expected to be low since it is the purpose of management to control the quality and quantity of herbage offered to the animals [7]. In addition, because data were collected from a single farm, little variation of herbage NV influential factors including soil type, soil fertility, fertilization and grazing policy have most likely contributed to the reduced variability of herbage NV samples.

The fact that CP was predicted with higher accuracy than DOMD, ME, NDF and ADF (Table 5) may partially reflect the incidence of the herbage sampling method used. Asner [37] describes that even when the leaf area index (LAI) is high (LAI > 5), the lower strata of a sward can influence the spectral signature of grass canopies. If nutrient content in the lower strata, consisting of tissue of low CP and high fiber content (thus, low digestibility and ME), had an influence on the spectral

measurements, but was not considered in the calibrations, then such mismatch may partially explain the lower performance of fiber, DOMD and ME models over the CP model. In this study, it was assumed that because herbage samples were collected at the pre-grazing stage, the influence of the lower strata on canopy reflectance would be minimal. However, it is not possible to assert if the lower strata had a significant influence on canopy spectra from the data collected in this research.

Differences in herbage mass and or/soil fertility among sampling plots might also help explain differences in NV that might not have been accounted for by the calibration models based on canopy spectra. For instance, the mechanistic link between herbage mass and CP and digestibility has recently been theorized by Lemaire and Belanger [14]. These authors describe that as plants increase in size, structural tissues, of low N concentration and low digestibility, and metabolic tissues, of high N concentration and high digestibility, also increase, with the former increasing more rapidly than the latter. The difference in accumulation rates of tissues of different NV results in a negative relationship between herbage mass and CP and digestibility, which is exacerbated if differences in soil N availability are present [14]. Although herbage samples in this study were collected at the pre-grazing stage, heterogeneity at the paddock level could have resulted in differences in mass and soil fertility at the sampling plot level. If differences in mass and soil fertility were true and not strongly reflected by canopy spectral measurements, then the predictive capability of empirical calibration models such as the ones developed in this study could potentially improve by the inclusion of these variables during the modelling process.

The technique used to acquire spectra in the field might also have contributed to predictive error. Characterizing the optical properties of herbage confined to the area set by the wooden quadrat may have not been adequate given the limited number and distribution of spectral measurements acquired within the sampling plot. The error associated with the determination of herbage NV of reference samples is also likely to have influenced the accuracy of the models. For instance, a lower error associated with the determination reference values for CP compared to NDF (Table 1) may partially explain the better predictions of CP over NDF by their respective canopy spectral models. The success of the NIRS calibration technique is dependent on good-quality reference data [11]. Thus, better predictions can be expected if models are built exclusively with wet chemistry determined reference data, as this is the standard procedure to determine the true value of nutrient content in animal feed samples [10].

### 4.3. Wavelength Contribution for Predicting Herbage Nutritive Value

The relationships between wavelengths in the far-SWIR region and VIP values of all herbage NV traits can be associated with the absorption of C-H bonds common in organic compounds that form fibers and proteins [49]. In accordance to this study, Clark and Lamb [54] described that the digestibility of the fibrous portion of plants is highly related with absorbance in the 2300 nm wavelength. Moreover, Curran [49] identifies that absorption in the range of 910 to 990 nm relate to oil and starch content, while absorptions between 1900 to 1980 nm relate to starch, cellulose and lignin, which can help explain the importance of wavelengths in these ranges on predicting contents of ADF, NDF, DOMD and ME found in this study. The concentration of chlorophyll–protein complexes, the major source of protein in vegetation, can explain the importance of the relationships between VIS and red-edge wavelengths and CP found in this research. Chlorophyll pigments were found to be good absorbers of electromagnetic energy in the visible region [49] and the photosynthetic activity of chlorophylls was also found to be highly responsive of reflectance in the red-edge wavelengths [49,55,56]. At the far end of the spectrum, the relationships between canopy wavelengths and CP can be attributed to the ability of the sensor to detect chemical bond activity linked to N. Curran [49] identifies that the wavebands centered at 2130, 2180 and 2300 nm are linked to the absorption mechanism of vibration of N-H and C-H stretch bonds in proteins.

Many of the wavelengths that were found to be important predictors of herbage NV from canopy spectra were also found important in other similar studies [19,20,32,34]. For instance,

visible wavelengths and wavelengths ranging from 2140 to 2300 nm were consistently important in the determination of CP [19,20,32,34], while spectra around 2220 to 2250 nm were consistently related to ADF [20,32]. Interestingly, in this study, a relatively large proportion of wavelengths in the NIR-plateau (800–1400 nm) were related to any of the NV traits compared to other studies. The NIR plateau is highly determinant of canopy health and structural features such as LAI and biomass [16] and has a limited to null relationship with the concentration of biochemicals in plant tissue [49]. It is possible that reflectance in wavelengths in this region is acting as a covariate of herbage NV in our models, resulting these wavelengths in an indirect measure of the NV trait. For instance, it would be expected that at the pre-grazing stage, a vigorous canopy of high LAI could be associated with a higher CP content than an unenergetic canopy of low LAI. Darvishzadeh et al. [57] identified that the LAI of grass canopies is strongly determined by reflectance at 1114 nm, because this wavelength has been related with CP in our modelling, then the explanation suggested above could be plausible. It is also possible that water absorption is acting as a confounding factor in the models, resulting in a relatively higher importance being attributed to NIR over SWIR wavelengths. Kumar et al. [58] describes that water absorption can overshadow biochemical features at wavelengths beyond 1400 nm and that this is the main reason for improved predictions from dried over fresh foliage in laboratory studies.

## 5. Conclusions

This study showed that proximal hyperspectral measurements of dairy herbage canopies were useful to predict the NV of the vertical portion of herbage that is available to the grazing cow. The PLS regression approach used in this research indicated that the relationships between the spectra and CP were stronger ($R^2 = 0.78$) than the relationships obtained between the spectra and DOMD, ME, NDF and ADF ($0.54 < R^2 < 0.67$). Being able to utilize proximal sensing for measuring the NV of the herbage available to the grazing cows in the field could lead to more efficient grazing management. Improved precision of herbage allocation at any single grazing event can lead to potential short- and long-term efficiency and productivity gains at the farm level.

**Author Contributions:** Conceptualization, F.N.D. and I.J.Y.; methodology, F.N.D.; software, F.N.D.; validation, F.N.D.; formal analysis, F.N.D. and N.L.-V.; investigation, F.N.D.; resources, I.J.Y., N.M.S. and N.L.-V.; data curation, F.N.D.; writing—original draft preparation, F.N.D.; writing—review and editing, N.L.-V., S.T.M., N.M.S., I.D. and I.J.Y.; visualization, F.N.D.; supervision, I.J.Y., N.M.S., S.T.M., N.L.-V. and I.D.; project administration, N.M.S. and I.J.Y.; funding acquisition, I.J.Y. and N.M.S. All authors have read and agreed to the published version of the manuscript.

**Funding:** This research received no external funding.

**Acknowledgments:** The authors would like to thank Eduardo Sandoval-Cruz for helping with collecting field data.

**Conflicts of Interest:** The authors declare no conflict of interest.

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
