# Peer review of "Using Proximal Hyperspectral Sensing to Predict Herbage Nutritive Value for Dairy Farming"

_agronomy, doi:10.3390/agronomy10111826_

Round 1

Reviewer 1 Report

Article: Using proximal hyperspectral sensing to predict herbage nutritive value for dairy farming (# 970206)

Manuscript review: reviewer’s comments

  1. The title is in accordance with the contents of the article.
  2. Introduction is sensible, systematic and well written (1 comment).

Line 48: Use unabbreviated terms (here and afterward) followed by abbreviations in brackets when mentioned first (refer to the journal list of standard/non-standard abbreviations).

  1. Material and methods are described systematically and in details (3 comments).

Line 106: Please, add the period used for average calculation of weather parameters.

Line 149: Is the temperature of 70 °C correct? For drying fresh herbage, the temperature of 60 °C is standardly applied (see your reference source).

I am not able to estimate method items under points 2.2. and 2.6. due to lack of experience considering the spectral measurements. I am also not familiar with the software packages used for statistical procedures. Therefore I can not estimate their selection correctness.

Line 223: Please, add “Metrics of” goodness …

  1. Results need to be improved significantly (10 comments).

Line 241: The caption of Table 3 is incorrect (?) (same as the caption of Table 2)

Line 247: These data are secondary for this research.

Line 261: Please, change “treatment” with “the pre-treatment”

Lines 261-263: The described characteristic in this text is not presented in Figure 1b (?) (please, see the comment on this figure below).

Lines 265-266: The legend of the curves is incorrect and uncomplete. Please, define black and green lines.

Figure 1b is unclear for me (probably defective). At secondary vertical axis the title (CV%) and units (0 to -200000) are not compatible (?).

Line 268: Please, amend the section title at point 3.2. to cover both sub-points (3.2.1. and 3.2.2.)

Line 281: Please, change “0.62” with “0.67”

Line 289: Please, change “Figure 1” with “Figure 2”

Lines 305-307: Why did you here use FDA for describing relevance of wavelength for CP determination? In the other mayor part of the section you used wavelength.

  1. Discussion is sound in all relevant aspects (1 comment).

Line 313: In the following sentence comparison is unclear. Similar values for ME, CP, DOMD, NDF and ADF… (similar to what?)

  1. Conclusion

Line 427: … (0.57 < R2 < 0.61). This range is not identified in the result section.

  1. References

Style, format and correctness of references have not been checked in details.

  1. Abstract (1 comment)

Line 22: … (0.54 < R2 < 0.67). I suggest putting it at the end of the sentence. Finally, the range is correct (same as in Table 5).

General comment

I would suggest revision pertaining to my specific comments before acceptance. To avoid trivial errors I would suggest the authors, especially the first one to be stricter at writing and self-editing a manuscript before submitting.

Reviewer 2 Report

This paper deals with attempts to estimates several forage quality parameters through measurement of spectral reflectance of sward canopy. This application interest of such a study is manifest and if successfull this kind of in situ determination can be advantageously incorporated within grazing management decision tool.

This work was performed  by using sophisticated statistics for relating "spectral measurements" with different forage quality traits determined on forage sampling by chemical analysis. These traits are the classical traits used in literature for analysing forage quality for herbivore feeding. As I am not an expert in statistic nor in spectral analysis, I cannot provide my own opinion on this part of the work. I hope an other reviewer would have this expertise for analysing more deeply the accuracy of the methodology. I would concentrate my review on the more biological interpretation of this work.

The approach used in this paper is a purely "black-box" one, based on statistical relationships between "spectral measurements and quality traits. I don' deny the predictive capacity of such an approach, but the generalization of this predictive capacity to a wider range of conditions-situations would be re-inforced if these "correlations" are also supported by some functional evidences. So my impression is that this paper is only concentred on statistical analysis, without any reference to more mechanistic model allowing understanding of quality trait dynamics during herbage growth process in order to enrich the discussion.

In recent review that were published in Agronomy, authors should find relevant references to allometric relationships linking quality trait to both plant growth and canopy structure. By referring to this models, it should be possible to provide more comprehensive interpretations for the statistical relationship obtained in this manuscript.

For example, the question to know if the use of herbage mass as co-variable for explaining the part of the variablity non explained by statistical relationship should be very relevant.  Authors identified well that repartition of N within canopy being not homogeneous...and reflectance being mainly sensible to the higher layers in the canopy... there is a distorsion between average CP concentration of forage and the CP concentration detected by reflectance. A clear analysis of this question is provided within this review and should be used for discussion.

So my conclusion is that this manuscript is an incomplete paper... The statistical approach must me completed and enriched with reference to "biological" and "ecophysiological" comprehensive models. By doing this effort I think authors should provide a more convincing conclusion and interpretation to their work. So I recommand "major revisions" for this interesting and promising manuscript.

Round 2

Reviewer 2 Report

I think that response of authors to my recommendations is positive even if "a minima". We could have expected that the introduction of "crop mass" as a co-variable for explaining the variability of forage nutritive value within the statistical analysis of data would have been a clear scientific added value to this paper. But I accept this "a minima" improvement because I realize that it would have represented a large quantity of work... I encourage authors to do this work for a following paper. So I think this paper could be now accepted for publication.